# Assessment of the Potential Flood Hazard and Risk in the Event of Disasters of Hydrotechnical Facilities—The Exemplary Case of Cracow (Poland)

**Jerzy Grela**

Engineering and Environment Department, MGGP S.A., Lea 112, 30-133 Cracow, Poland; jerzy.grela@mggp.com.pl

**Abstract:** The article presents an analysis of the potential consequences for Cracow of failure of dams located in the Vistula catchment area upstream of the city. They have been compared with the effects of a flood with a probability of 0.2%. The estimation of losses and damages was made on the basis of the results of model studies and flood hazard and flood risk assessments carried out as part of the analyses of the second planning cycle of the implementation of the Floods Directive in Poland. The MIKE 11 model was used to simulate flows in riverbeds and its Dambreak module is used to simulate dam failure. An adjustment of the loss index for Cracow's residential areas was proposed. Some elements of the described methodology can be used to estimate flood losses in large cities. They can also be applied to cities that are not at risk of dam disasters.

**Keywords:** dam disaster; flood risk; flood losses; residential area

## 1. Introduction

The last few decades have seen a worldwide occurrence of flash floods causing enormous losses, especially in urban areas. The floods also affect cities of great importance to European and global culture (e.g., Florence and Venice 1966, Ayutthaya 1995, Wrocław 1997, Prague, Grimma 2002). Literature offers examples of special studies dedicated to estimating potential damage and loss in such cities as Florence [1] and Cologne [2]. Plans are also being developed for the preservation of historic cities, e.g., Cologne [2], Venice [3], or Grimma [4]. Many international organizations have recognized the problem of protection of cultural assets for multiple years, as evidenced by, for example, the Kyoto Declaration of 2005 on the protection of cultural assets, historic sites, and their surroundings in the event of a disaster [5], or the joint United Nations Educational, Scientific, and Cultural Organization (UNESCO), International Center for the Study of the Preservation and Restoration of Cultural Property (ICCROM), International Council on Monuments and Sites (ICOMOS), and International Union for Conservation of Nature (IUCN) manual on world heritage disaster risk management [6].

In an era of undeniable climate change, natural floods will occur, compounded by anthropogenic human activity. Hence, in 2007, the European Union countries prepared the Floods Directive [7], a document indicating framework directions for Member States to minimize the impact of these events. The flood scenarios recommended in this document include not only natural floods, but also extreme scenarios resulting from disasters of hydrotechnical facilities (embankments, dams).

Many publications have been devoted to dam disasters around the world. Some of them are descriptions of individual events—actual [8] or hypothetical [9,10]—while others evaluate and classify the causes of these disasters based on an analysis of multiple cases [11–13]. The main causes include:

- Environmental factors—excessive rainfall (Shimentau and Banqiao, China, 1975, Morvi, India, 1979, Malpasset, France, 1959, Way Ela, Indonesia, 2013), landslides into a reservoir (Vajont, Italy, 1963), earthquakes (Huaraz, Peru, 1941)
- Design errors (Madrid, Spain, 1905, Gleno, Italy, 1923, Malpasset, France, 1959, Austin, USA, 1911 and 1941) and construction errors (Bila Desna, Czech Republic, 1916),
- Conscious human activity (Dnieper Reservoir, USSR, 1941).

A significant number of publications deal with numerical tools used to simulate the failure of concrete or earth dams. The most popular models in the world today are as follows:

- HEC-RAS (a computer program for one- and two-dimensional hydraulic calculations in natural and artificial bed networks, designed and developed by the US Army Corps of Engineers); applications of the one-dimensional model include the Asa Dam in Nigeria [14], the Mosul and Chaq Chaq dams in Iraq [15,16], the two-dimensional model was used for the Mulacorreal Dam in Ecuador [17] and Way Ela in Indonesia [18], among others,
- MIKE (a family of one- and two-dimensional models for hydraulic computations in natural artificial channels and drainage networks, initiated and developed by DHI); applications of Mike 11 models include simulations of the failure of the Besko concrete dam in Poland [10], the Mike Flood model was used for the Dark Dring dam in Vietnam [19],
- TELEMAC-2D (for simulating free surface flows in two dimensions of horizontal space; solves Saint-Venant equations using finite element or finite volume methods and the triangular element computational grid, initiated and developed by EDF); this model was used, for example, to analyze the potential disaster of the Mexa and Bougous dam cascade in Algeria [9].

A group of publications discuss the specific problem of determining breach and catastrophic wave parameters in earth dam failures. These parameters are selected using various formulas proposed by authors such as Froehlich [20], Macdonald and Langridge-Monopolis [21], Von Thun and Gillette [22], the U.S. Bureau of Reclamation [23], Pierce M. W.; Thornton C. I. [24], Singh, K. P.; and Snorrason, A. [25,26], Xu, Y., Zhang, L. M. [27]. Some works discuss the evaluation of these formulas with specific examples, indicating those that seem to be more effective in their particular cases, e.g., [15,16,28,29].

The present study refers to the above-mentioned aspects, as it concerns the estimation of the effects of potential dam disasters in the area of Cracow, one of the important centers of Polish and European culture, using common numerical tools. Modifications to the methodology of estimating losses in residential areas used in Poland were also proposed, taking into account the unique characteristics of a large urban center.

## 2. Materials and Methods

As part of the second planning cycle (2016–2022) for the implementation of the Floods Directive, the following scenarios were performed in Poland in accordance with the Water Law Act:

For natural floods

- Scenario I—areas with a low probability of flooding of 0.2% (once every 500 years);
- Scenario II—areas with a medium probability of flooding of 1% (once every 100 years);
- Scenario III—areas with a high probability of flooding of 10% (once every 10 years);

And hydraulic facility disasters

- Scenario IV delineating areas prone to flooding in the event of embankment destruction or damage (delineated for a flow with a probability of 1%)—a scenario of total embankment destruction;
- BP scenario designating areas prone to flooding in the event of destruction or damage to the damming structure.

The methodological assumptions underpinning the development of scenarios I–IV are contained in paper [30], while for the BP scenario, they are contained in paper [31].

The remainder of this paper includes an analysis of the consequences of flooding that may be caused by the failure of structures built to protect against natural floods. This is because, paradoxically, a structure which is intended to protect against flooding may, in the event of failure, generate a flood phenomenon that is just as dangerous as or even more dangerous than a natural wave.

This is quite often the case with embankments. Embankments breaking cause great damage, because the areas outside the embankment are usually densely developed. This is due to a false sense of security in the protected areas (the scenario of an embankment break in Cracow is discussed in paper [32]).

A dam failure can also cause massive flooding even at a considerable distance from the failure site. In Poland, the consequences of failure of 26 dams, including 22 earth dams and 4 concrete dams, were analyzed between 2019 and 2022 [33].

The modeling of the failure process and its effects in the river basin is described by the methodology [31]. Its key assumptions are listed below:

- The flood risk area resulting from a disaster or dam failure is determined by modeling transient movement in river channels,
- The modeling uses the tools of the Mike family, by DHI [34],
- The reservoir model is an integral part of the hydraulic model; the description of the reservoir bowl geometry is based on the reservoir basin capacity curve while the reservoir head dam has been implemented as a wide crest overfall with parameters corresponding to the actual characteristics of the facility,
- The operation of the reservoir was described in accordance with the flood water management manual and the adopted assumptions regarding the failure of the spillway equipment—the control policy was implemented in the 1D model using the CONTROL STRUCTURE function,
- The dam failure parameters were implemented in the computational structure responsible for representing the disaster—DAMBREAK STRUCTURE in the 1D model (Mike 11),
- The significant damage parameter—the width of the breach in the earth dams—was determined as the arithmetic mean calculated using three formulas, based on the Froehlich [20], Macdonald and Langridge-Monopolis [21], as well as Von Thun and Gillette [22] formulas,
- Models developed for the needs of MZP and MRP for river floods (scenarios I–IV) [35] were used for hydrodynamic modeling of the valley downstream of the dam, calculations were based on 1D models (Mike 11),
- The hydrogram of the inflow to the reservoir, which corresponds to a wave with a culmination equal to the control flow for the dam, was assumed as the upper boundary condition in the developed models. The lower boundary condition, which concluded the model, was the water level hydrograph or flow rate curve. The internal boundary conditions (concentrated and distributed lateral inflows) were assumed to be identical to those in scenario I (Q0.2%),
- The scenario of destruction of or damage to damming structures assumes the variant with the largest ensuing flood risk zone area (OZP) as its basis.

It should be noted here that the methodological assumptions adopted in Poland to determine flood hazard and flood risk maps (MZP and MRP) for natural floods [30] and dam disasters [31] do not take into account the phenomena of sediment movement, erosion and sedimentation occurring in the channel. As a consequence, these phenomena were not included in the hydraulic modeling. Some authors indicate that erosion caused by a flash flood can cause a significant lowering of the channel and thus the water level by up to 5 m (example of the Qiantang River, Xie et al. [36–39]). Phenomena of such a scale do not occur in Poland, they are only local in mountainous areas, where the destructive force is high with large slopes of the channel and high water velocity. The Vistula in Krakow is already a lowland river, at a considerable distance of over 100 km from the site of a potential disaster,

with a bed additionally stabilized by three water barrages, so there is no risk of a significant deepening of the bed and significant lowering of the water level.

The effects of flood disasters are considered in terms of flood hazard and flood risk. Maps (MZP and MRP) are the means of visualizing the extent of the flood hazard and risk. Cartographic versions of these maps are developed separately for each flood scenario.

Flood hazard is defined as the most generally identified floodplain areas together with analyses of the depth and possibly the velocity of the moving water. It is therefore a picture which indicates where the water may end up as a result of a disaster, and what the depth (and possibly velocity) will be at any point in that area. By determining the limit values of these quantities, more and less dangerous places within the floodplain can be identified.

The flood variant involving the destruction of or damage to a damming structure (the BP scenario, according to the nomenclature adopted in the Floods Directive due to origin, is an A15 flood) is modeled for the control flow at the inflow to the reservoir, whereas on all lateral inflows downstream of the reservoir, it is as in scenario I, i.e., for Q0.2%. Disaster variants are constructed for each facility—puncture or overflow for earth dams and damage to one or more overfall sections for concrete dams. Disaster diagrams for a single facility and cascaded facilities are shown in Figure 1.

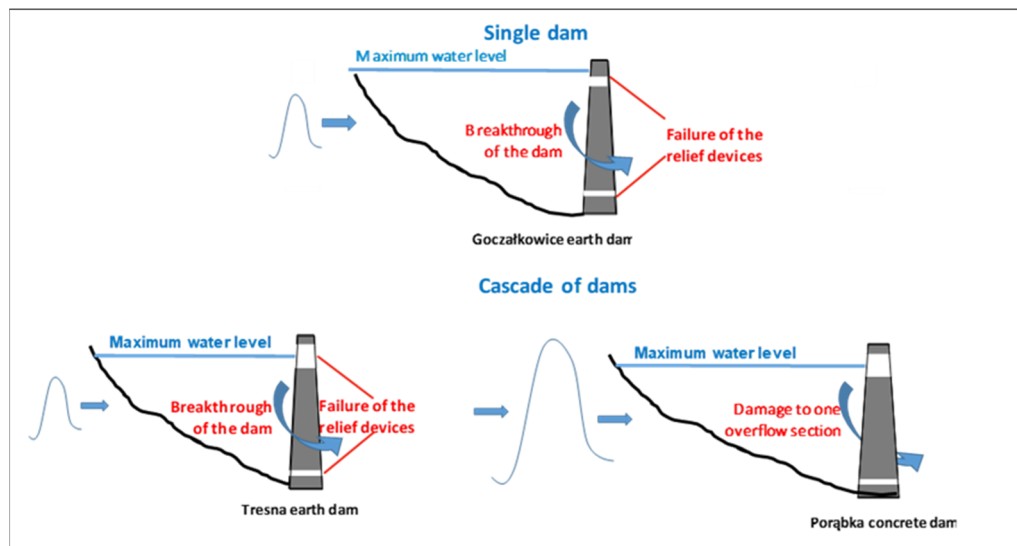

**Figure 1.** Disaster patterns for a single dam and for dams located in a cascade (the white fragments of the dams are a schematic representation of the overflow and discharge devices, and the blue curves and arrows are the direction of the wave inflow to the reservoir and its symbolic shape).

Two BP disaster scenarios for the Goczałkowice and Tresna and Porąbka dams were selected for the Cracow analysis and compared with scenario I. The parameters of these dams are given in Table 1.

The dam disaster scenario may be realized in Cracow as a result of destruction or damage to several reservoir structures located upstream of the city. For the selected three structures, Goczałkowice, Tresna (including Porąbka), disaster variants were developed and calculations for each structure indicated which variant is the most unfavorable from the point of view of the floodplain (described in Table 2). In the case of the Tresna dam, it was assumed that its disaster would also cause a disaster at the Porąbka dam, located downstream in the Soła cascade, so in this case the disaster would affect two structures simultaneously (Figure 1).

A schematic diagram of the location of the analyzed dam structures upstream of Cracow whose destruction or damage puts the city at risk is shown in Figure 2.

**Table 1.** Parameters of the analyzed damming structures.

| No. | Name of Reservoir | River | River Kilometer | Distance from Cracow (km) | Structure Category | Year of Construction | Type of Dam | Flood Reserve (m³ Million) | Volume of Water at MaxPP (m³ Million) | Water Volume at Dam Crest Level (m³ Million) |
|---|---|---|---|---|---|---|---|---|---|---|
| 1 | 2 | 3 | 4 | 5 | 6 | 7 | 8 | 9 | 10 | 11 |
| 1 | Goczałkowice | Vistula River | 43.1 | 115.0 | I | 1955 | earth | 43.18 | 161.25 | 217.0 |
| 2 | Tresna | Soła | 40.2 | 110.3 | I | 1967 | earth | 30.64 | 92.70 | 121.5 |
| 3 | Porąbka | Soła | 32.3 | 102.4 | I | 1936 | concrete | 4.50 | 26.54 | 31.84 |
| | | | | | | | TOTAL | 138.42 | 441.33 | 556.15 |

**Table 2.** Selected variants of damming structure disasters.

| No. | Name of Reservoir | Scenario Description | Width of Breach—Puncture | | | average (m) | Culmination of Inflow at the Time of the Disaster (m³/s). | Outflow Culmination at the Time of the Disaster (m³/s). | Disaster Extent (km) |
|---|---|---|---|---|---|---|---|---|---|
| | | | Based on Froehlich Formula (m) | Based on Macdonald and Langridge-Monopolis Formula (m). | Based on Von Thun and Gillette Formula (m). | | | | |
| 1 | 2 | 3 | 4 | 5 | 6 | 7 | 8 | 9 | 10 |
| 1 | Goczałkowice | Damage to the Goczałkowice reservoir dam as a result of a hydraulic breakdown of the dam body | 124 | 902 | 91 | 369 | 1618 ($Q_{0.05\%}$) | 7660 | 171.35 |
| 2 | Tresna+ Porąbka | Damage to the Tresna reservoir dam as a result of hydraulic breakdown of the dam body combined with the destruction of one overfall section of the Porąbka concrete dam | 106 | 372 | 105 | 194 | 3052 ($Q_{0.02\%}$) | 14,210 | 186.55 |

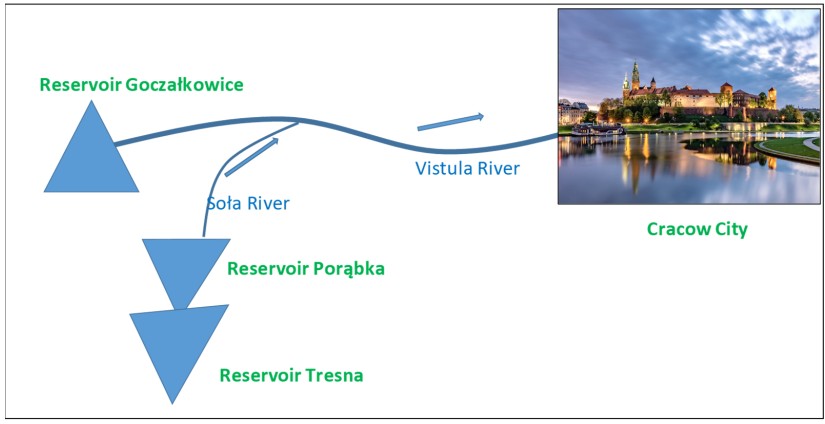

**Figure 2.** Location of analyzed damming structures upstream of Cracow.

The flood risk shows the effect of the threat in terms of the number of inhabitants at risk as well as facilities and areas with a diversified development structure. Where possible, risk is expressed in monetary units, expressing the magnitude of potential flood losses. The risk is strictly a function of the hazard; it generally grows (although not for all development categories) as the depth and velocity of water in the floodplain increases.

This risk concerns the threat to the population, facilities of special social importance, and particular land use categories. The threat to the population is defined by the number of people at risk of flooding, facilities of special social interest by type and number (but excluding occupants), while land use is defined by the area at risk and the value of the losses in monetary units.

The exact number of inhabitants of a large city at risk cannot be precisely determined at the time of a flood. This is determined both by demographic changes over shorter and longer timeframes and by seasonal migrations related to tourism and the academic year, as well as by population mobility over the daily cycle. Only estimates of permanent residents in potential floodplain areas are possible, but even these will remain just an approximation.

In order to estimate the number of occupants in buildings located in the flood risk area as reliably as possible, the methodology [30] lists the following calculation groups: single-family residential buildings, buildings with two dwellings, and multi-family buildings. Both the information on the nature of the building and its geometry are determined from the national database of structures (BDOT10k) from the 'Buildings, structures and facilities' layer. The first step is to calculate the number of households in each municipality. Based on buildings extracted from BDOT10k, each single-family residential building is assigned to one household, while buildings with two dwellings are assigned to two respective households. To estimate the number of households in multi-family buildings, a formula is used from which the number of dwellings per story is first calculated based on the BDOT building footprint, the number of stories (BDOT), and the average floor area of one dwelling in m$^2$. Then, by multiplying the result obtained by the number of stories in the building, the estimated number of dwellings is calculated (CSO data for municipalities).

Based on the above data for municipalities, buildings located in the flood risk area are assigned their number of inhabitants (LM) by multiplying the number of households in the building by the average number of people per dwelling in the municipality (the average number of people dwelling in the municipality is published on the CSO website at the Local Data Bank). The obtained result, when rounded to the nearest whole number, renders an estimate of the number of residents in a given building.

Risk to buildings of special social importance is distinguished based on the purposes they serve. The level of flood risk for such facilities on risk maps is detailed by a color-based representation of the maximum flood level for the facility, with a limiting depth of 2 m.

The method of calculating the value of potential flood losses (quantitative and monetary) has been defined for individual land use categories in a methodological study based on the methodology applied in the German state of Brandenburg [40]. In the current planning cycle, some assumptions have been modified and adapted for Polish conditions, using national data [30,41,42].

The calculation of potential flood loss values is performed for seven land use classes:

- Class 1—residential areas;
- Class 2—industrial areas;
- Class 3—communication areas;
- Class 4—forests;
- Class 5—recreational and leisure areas;
- Class 6—arable land and permanent crops;
- Class 7—grassland.

For other land and for surface water, potential flood losses are not calculated due to the lack of use or insignificant development of these areas.

For classes: residential areas, industrial areas, communication areas, the value of losses is obtained by multiplying the property value for each class by the asset loss rate depending on water depth (Figure 3).

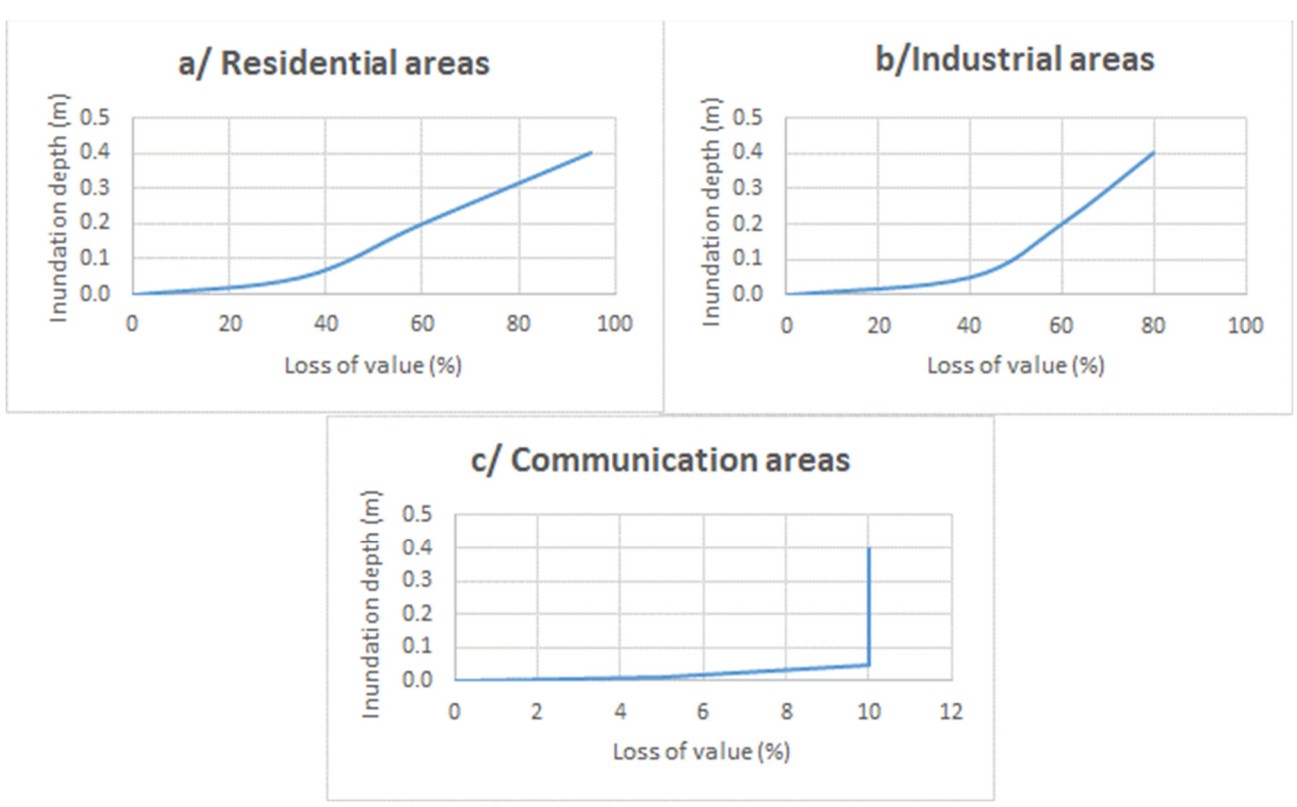

**Figure 3.** Loss of property value for a given use class depending on water depth.

Property value for residential areas varies according to the voivodship's GDP to national GDP ratio, for industrial areas, the measure of direct losses is the gross value of fixed assets in individual voivodships relative to the area of industrial areas, for the following two: agricultural land and permanent crops, as well as grassland depending on the volume of the voivodship's agricultural output.

For classes: forests and recreational and leisure areas, the value of losses of endangered assets does not depend on water depth. For these classes and communication areas, the loss rates are constant throughout the country.

The direct losses calculated using the above method include indirect losses (does not apply to communication, forest, recreation, and leisure areas), defined as a % of direct losses. The value of indirect losses is assumed to be:

- Densely built-up residential buildings
- Housing development with loose buildings
- Industrial areas
- Arable land and permanent crops
- Grassland

- 80%
- 40%
- 80%,
- 20%,
- 30%.

Table 3 shows the (total direct and indirect) loss rates of the Małopolska voivodship area which were adopted in the methodology documentation [29]. The methodology makes it possible to estimate potential flood losses for each flood-prone area and for each probability of occurrence. As the purpose of this article is to estimate losses in the Cracow area, the use of voivodship rates for residential areas is subject to an error of underestimation. This was pointed out in the paper [32], where the rate used for Cracow

was the same as for the Małopolska voivodship for the preliminary assessment of the magnitude of losses in Cracow.

**Table 3.** Per-unit loss rates for the Małopolska voivodship and Cracow (2016 price level).

| Class | Type of Use | Unit Loss Rates (PLN/m²) | |
|---|---|---|---|
| | | Małopolska Voivodship | Cracow Municipality |
| 1 | 2 | 3 | 4 |
| 1 | residential areas | 514.05 | 2619.96 |
| 2 | industrial areas | 1028.11 | 1028.11 |
| 3 | communication areas | 717.00 | 717.00 |
| 4 | forests | 0.04 | 0.04 |
| 5 | recreational and leisure areas | 8.00 | 8.00 |
| 6 | arable land and permanent crops | 0.40 | 0.40 |
| 7 | grassland | 0.08 | 0.08 |

For the purposes of this study, the rate for residential areas (related to GDP) was converted to city level based on the scheme provided in methodological documents [30,41,42]. This scheme used for calculations at voivodship level was modified for the city of Cracow as shown in Figure 4.

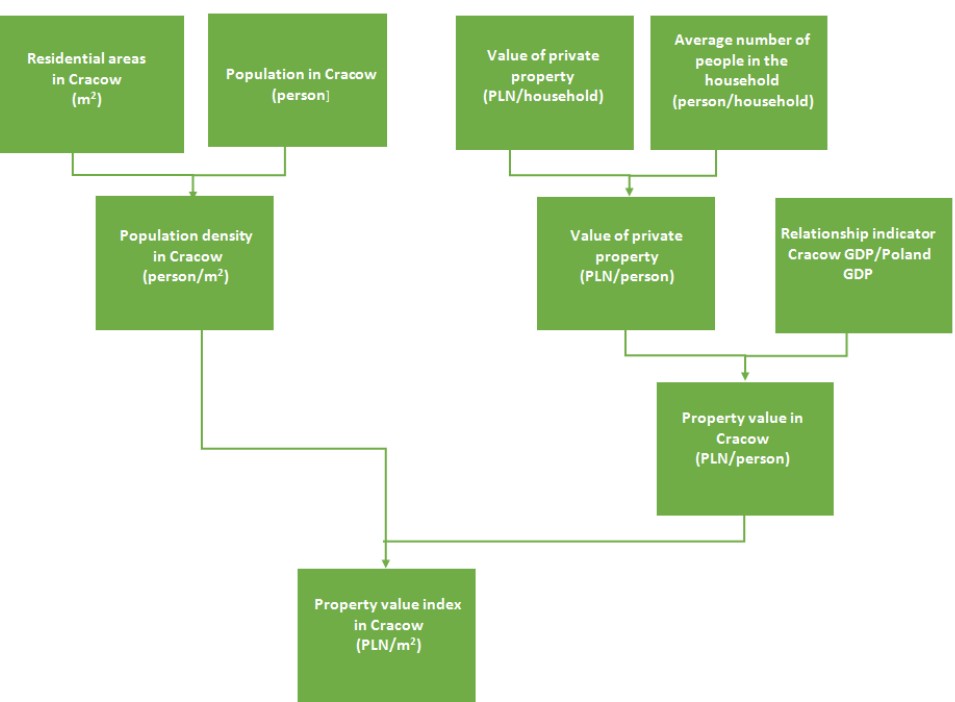

**Figure 4.** Modified scheme for determining the property value rate in residential areas in Cracow.

When calculating the value of the loss rate for residential development for Cracow, the relevant data for the city from 2016 [43–45] were adopted to keep the data in line with the NBP study on household wealth in Poland [46]. Based on these documents, the following conditions were adopted for Cracow:

- Median net household wealth in Poland—PLN 293,000
- Average number of inhabitants per household—2.04,
- Cracow's GDP index 63% higher than the national average GDP for Poland,
- Surface area of residential development in Cracow equal to 13.67% of the city area (44.62 km² out of 326 km²), scaled up based on the methodology [29] to include commercial development areas 7.28% of the city area (23.77 km²)

- Population of 765,320 people.
- Calculation process:
- Property value per person for median net worth 293,000/2.04 = PLN 143,627.45/person
- Raised by ratio of Cracow's GDP to national GDP 143,627.45 × 1.63 = PLN 234,112.74/person
- Population density in residential areas 765,320/68.387 = 11,191 people/km$^2$ = 0.011191 person/m$^2$
- Loss rate for residential areas in Cracow 234,112.74 x 0.011191 = PLN 2619.96/m$^2$

The resulting value of the loss index for residential areas in Cracow is PLN 2619.96/m$^2$, i.e., approx. €582.2/m$^2$ (assumed at 1€ = 4.5 PLN).

Due to the lack of similar statistical data broken down by communes, it is currently not possible to calculate Cracow's value of loss rates for other types of land use. Therefore, the same rates were adopted for them as for the Małopolska voivodship (Table 3).

Based on the database of spatial flood risk maps, it is possible to calculate the overall values of potential (direct and indirect) flood losses.

The overall value of potential losses for use classes 1–3 is expressed by the following equation:

$$Sp_i = \sum_{j=1}^{4} Sp_{ij} \times A_i \; for \; i = 1 \; and \; 2 \tag{1}$$

$$Sp_i = \sum_{j=1}^{2} Sp_{ij} \times A_i \; for \; i = 3$$

where

$Sp_i$ means overall values of potential losses for a given class $i$ (PLN);

$Sp_{ij}$ means the value of potential per-unit losses for class $i$ and depth interval $j$ (PLN/m$^2$);

$A_i$ means the area occupied by a given class $i$ (m$^2$), and the overall value of potential losses for use classes 4–7 is expressed by the equation:

$$Sp_i = St_i \cdot A_i \; for \; i = 4 \ldots 7 \tag{2}$$

where

$Sp_i$ means overall values of potential losses for a given class $i$ (PLN);

$St_i$ means the value of potential per-unit losses for class $i$ (PLN/m$^2$);

$A_i$ means the area occupied by a given class $i$ (m$^2$).

## 3. Study Area

Cracow is a large city in Poland, located on the Vistula River, which has seen dynamic development since World War II, and the capital of the Małopolska voivodship (Figure 5). It has an area of 326 km$^2$, is divided into 18 subdivisions (districts), is inhabited by more than 750,000 people, and has the largest accumulation of unique monuments in Poland. At the same time, it is a city at serious risk of flooding from the Vistula River and its tributaries, as evidenced by the fact that for hundreds of years the area of the present-day Cracow has been repeatedly hit by flood inundations, often causing death and damage to urban infrastructure and private property.

An overview of flood events in the Cracow area over the centuries is given in the monograph by A. Bielański [47]. It contains approximate floods of the then-Cracow area during the biggest historical flood of 1813, whose culmination was estimated at 3300 m$^3$/s (Figure 6).

A serious discussion on the protection of the city against floods has been going on since the beginning of the 20th century. As a result, specific investment measures were taken to reduce the threat from the Vistula (which remains the main source of danger for the city) and some tributaries. The bed of one of the Vistula's tributaries, the Rudawa, was relocated, and embankments and protective walls were built along the Vistula riverbed. In 1936, the first concrete dam was built upstream of Cracow–Porąbka, with a flood reserve of 4.5 million

m$^3$. In the period after World War II, further reservoirs were built: Goczałkowice (1956), Tresna (1967) and Świnna Poręba (2017), with flood reserves to retain about 134 million m$^3$ of water on the Mała Wisła, Soła and Skawa rivers. The construction of polders along the Vistula riverbed (upstream of Cracow), designed to retain a further 50.5 million m$^3$ of water, is currently being planned.

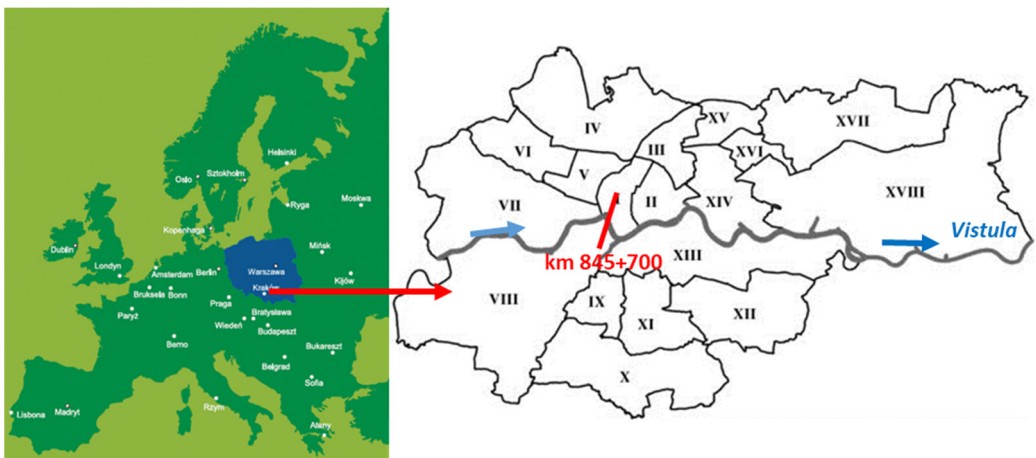

**Figure 5.** Cracow against the background of Europe and its administrative division into districts.

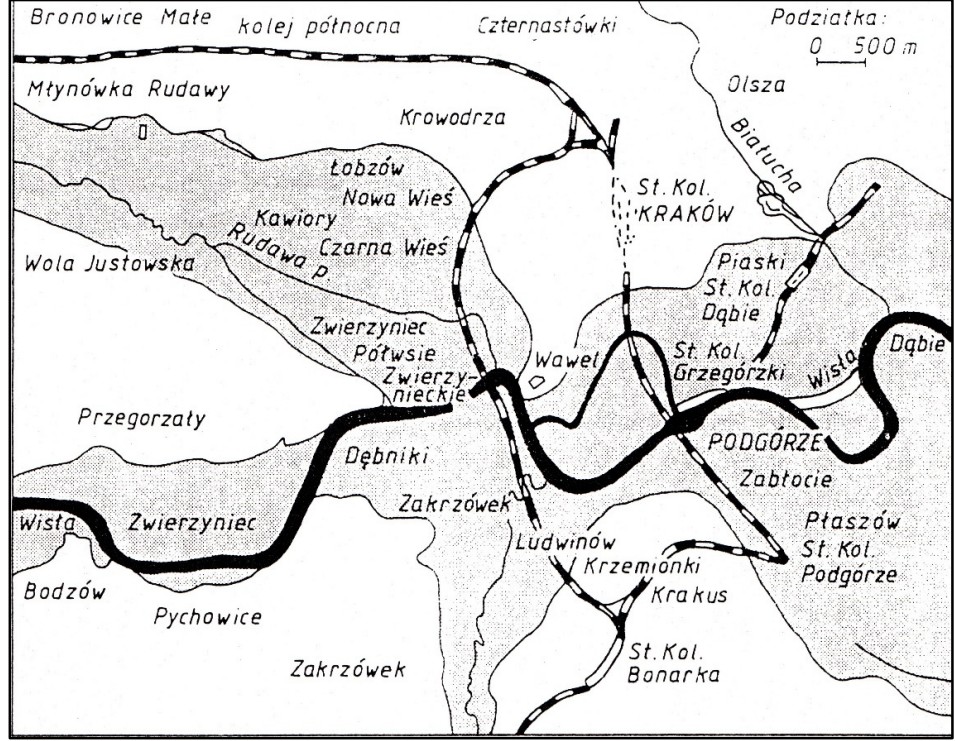

**Figure 6.** An overview plan of the extent of the flood water flood in Krakow from 1813 (names of housing estates and districts in Polish, have no equivalent in English), according to [47].

The work is expected to begin in the next planning cycle as part of the implementation of the Floods Directive between 2022 and 2027.

Despite many efforts over the last 100 years, the level of flood safety in Cracow is not satisfactory. The current embankments in Cracow (without the height margin required by the relevant regulations) provide for a flood wave with a probability of occurrence of approx. 0.4% and with an intensity of 2970 m$^3$/s at the point with the largest embankment

height deficit. On the other hand, taking into account the required elevation above the design and check water for class I embankments, we can only discuss securing Cracow against water with a probability of occurrence of 1% and a flow of 2330 m$^3$/s.

It should also be noted here that the present height of the embankments in Cracow is the result of a compromise between hydraulic engineering specialists and urban planners and cannot be further increased due to the landscape values of the Vistula valley near the Wawel Hill (km 845 + 700 of the Vistula River). The conditions of the historic center of Cracow, its protection as a UNESCO World Heritage Site in the context of flood protection, and the uncoordinated expansion of urbanized areas are discussed in more detail in the paper [48].

## 4. Results

The analysis of the threat and risk of catastrophic floods in Cracow was carried out for the whole city as well as for the individual 18 subsidiary districts.

### 4.1. Flood Risk from Disasters of Hydraulic Structures in Cracow

Below are examples of a selected sheet of the flood hazard map with water depth which presents the flood hazard areas with the distinction of four water depth zones (with limit values of 0.5 m; 2 m; 4 m) for catastrophic scenarios in the central part of Cracow (Figure 7a–c, the symbols used on the maps are explained in Table 4). Note the similarity of the flood extent with the historical flood of 1813 (Figure 6).

**Table 4.** Description of the map symbols in English.

| Symbol | Description of the Map Symbols in English |
|---|---|
| ▽ 72,56 | maximum water level |
| ● 75,15 | top of flood embankment elevation |
| ● 50 | chainage |
| ▭ | flood hazard area |
| | water depth h ≤ 0.5 (m) |
| | water depth 0.5 < h ≤ 2.0 (m) |
| | water depth 2.0 < h ≤ 4.0 (m) |
| | water depth h > 4.0 (m) |
| | watercourses and canals |
| | surface water |
| | flood embankment |
| | side dam |
| | dam |

**Table 4.** *Cont.*

| Symbol | Description of the Map Symbols in English |
|:---:|:---:|
| | location of dam failure |
| | commune boundary |
| | poviat boundary |
| | voivodeship boundary |
| | country border |

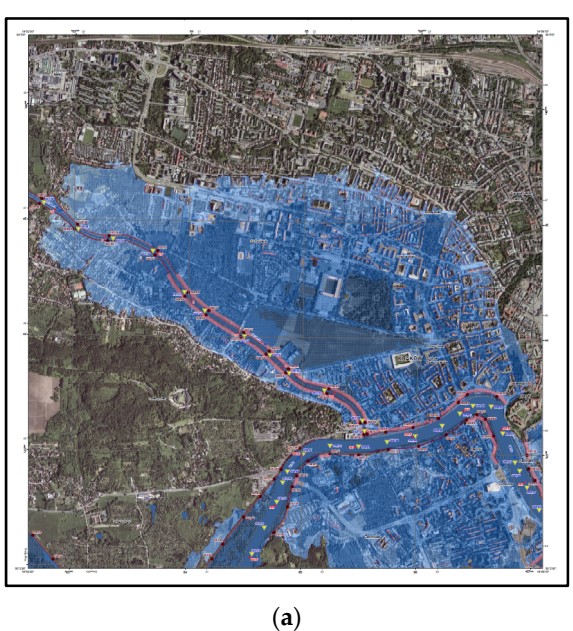

(**a**)

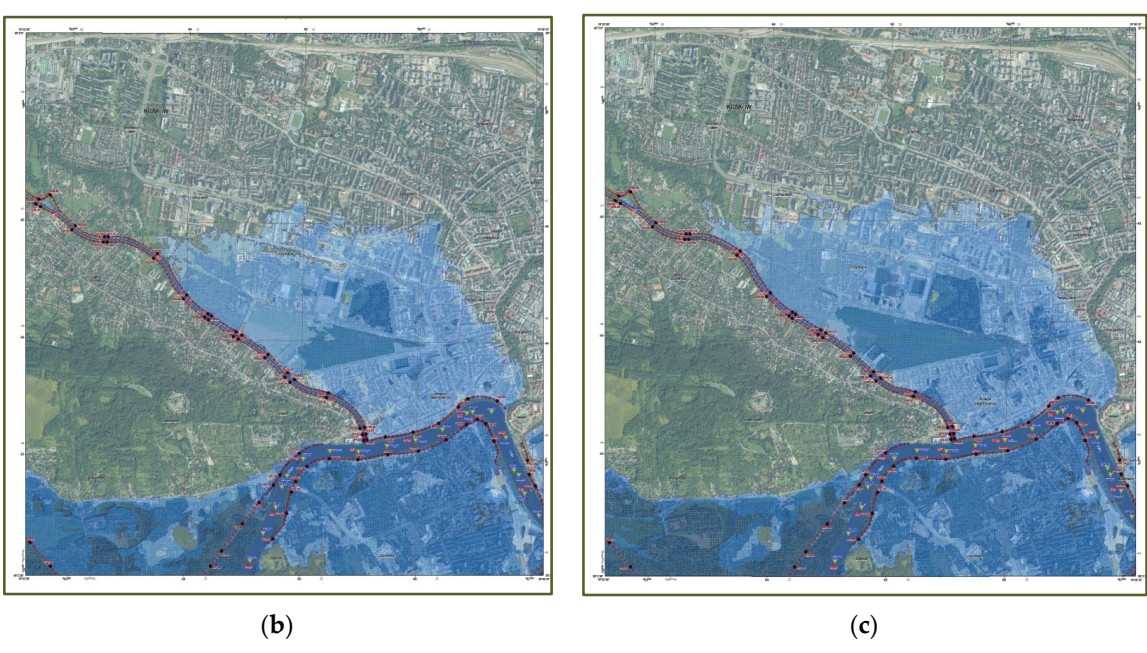

(**b**)                                     (**c**)

**Figure 7.** Flood hazard map with water depths (part of the area of Cracow). (**a**) Effects of a flood with a probability of Q0.2% in the Cracow inner-city area; (**b**) effects of the Goczałkowice dam disaster in the Cracow inner-city area; (**c**)effects of Tresna and Porąbka dam disasters in the Cracow inner-city area.

The data of (historical and probable) catastrophic floods were assessed in the profile of the Vistula River in the central part of the city [35,47,49] and are presented in Table 5 together with the parameters of culmination of waves originating from disasters of hydro-technical structures and resulting from hydraulic calculations with the MIKE11 model [33]. The visualization of the flood levels is shown in the schematic cross-section of the Vistula River at km 845 + 700 (Figure 8).

**Table 5.** List of elevation and culmination values of historic, probable, and catastrophic floods for the Cracow area (km 845 + 700 of the Vistula River).

| No | Type of Flood | Culmination | |
|---|---|---|---|
| | | Flow (m³/s) | Elevation (m a.s.l.) |
| | *Historical Floods (year)* | | |
| 1 | 1813 | 3300 | 206.71 |
| 2 | 2010 | 2476 | 204.65 |
| 3 | 1970 | 2300 | 204.21 |
| 4 | 1903 | 2250 | 204.09 |
| 5 | 1997 | 2100 | 203.71 |
| | *Probable Floods (p%)* | | |
| 6 | Q0.1% | 3870 | 208.13 |
| 7 | Q0.2% (scenario I) | 3224 | 206.67 |
| 8 | Q1% (scenario II) | 2411 | 204.54 |
| 9 | Q10% (scenario III) | 1336 | 201.87 |
| 10 | Q50% | 680 | 200.58 |
| | *Flooding Resulting from Disasters of Hydraulic Structures* | | |
| 11 | Embankment break (scenario IV) | 2411 | 205.15 |
| 12 | Goczałkowice dam disaster (BP scenario) | 3578 | 206.81 |
| 13 | Tresna and Porąbka dam disaster (BP scenario) | 3615 | 206.88 |

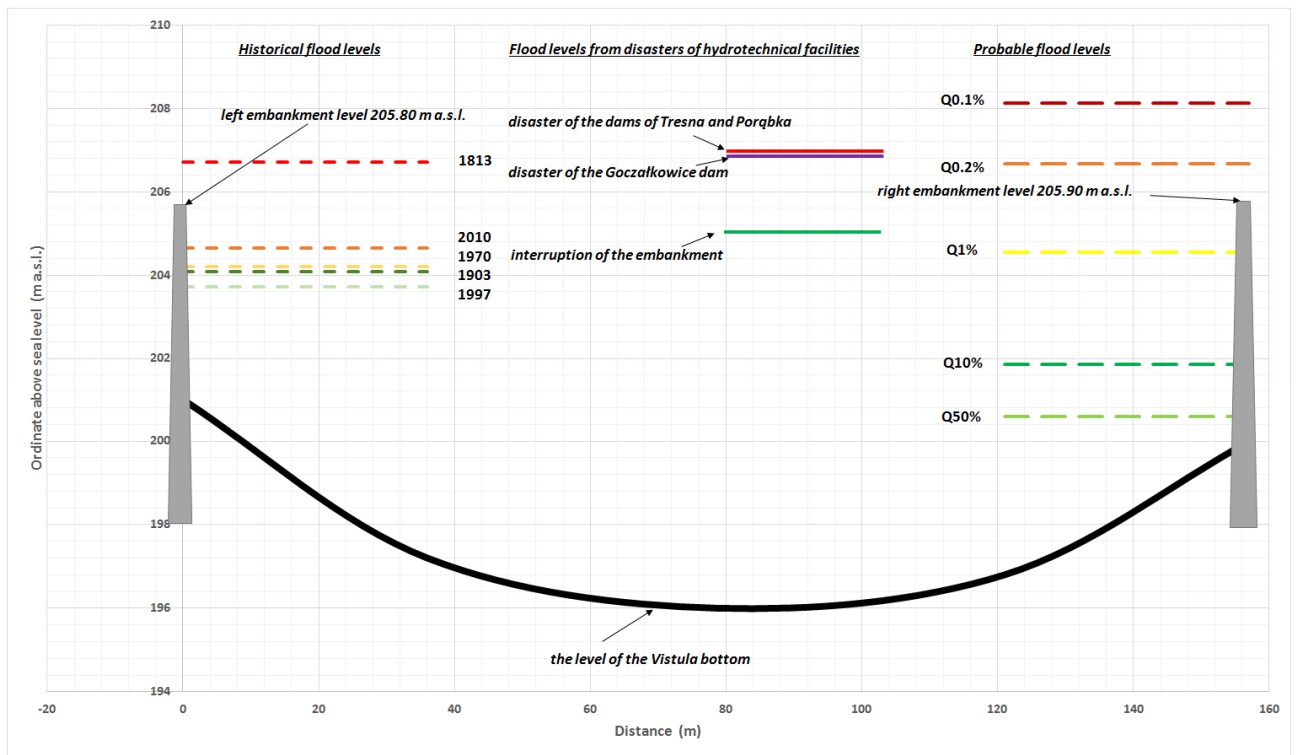

**Figure 8.** Flood level due to disasters of hydraulic structures in the central part of Cracow (diagram in the cross-section at km 845 + 700 of the Vistula River) against historical and probable floods.

Table 6 lists flood inundation areas, which enable an assessment of the scale of hazard of the various scenarios, their hierarchy and, in particular, an indication of which scenario is potentially the most dangerous for the city.

**Table 6.** List of flood inundations in Cracow for the analyzed flood scenarios, broken down by subsidiary district.

| Administrative Unit | Unit Area (ha) | Scenario BP Dam Disaster of Goczałkowice | | Scenario BP Dam Disaster of Tresna and Porąbka | | Scenario I Q0.2% | |
|---|---|---|---|---|---|---|---|
| | | Total Area of the Flood Lagoon (ha) | Percentage of the Unit Area (%) | Total Area of the Flood Lagoon (ha) | Percentage of the Unit Area (%) | Total Area of the Flood Lagoon (ha) | Percentage of the Unit Area (%) |
| 1 | 2 | 7 | 8 | 9 | 10 | 11 | 12 |
| District I | 556.05 | 160.12 | 28.80 | 174.54 | 31.39 | 122.21 | 21.98 |
| District II | 583.79 | 324.70 | 55.62 | 327.20 | 56.05 | 260.74 | 44.66 |
| District III | 642.98 | 2.62 | 0.41 | 2.63 | 0.41 | 1.92 | 0.30 |
| District IV | 2338.85 | 8.77 | 0.37 | 8.77 | 0.37 | 16.97 | 0.73 |
| District V | 561.18 | 148.89 | 26.53 | 189.50 | 33.77 | 226.36 | 40.34 |
| District VI | 954.71 | 5.78 | 0.61 | 6.37 | 0.67 | 45.23 | 4.74 |
| District VII | 2869.31 | 479.36 | 16.71 | 490.70 | 17.10 | 329.80 | 11.49 |
| District VIII | 4612.80 | 993.90 | 21.55 | 1155.66 | 25.05 | 424.82 | 9.21 |
| District IX | 540.81 | 6.56 | 1.21 | 6.56 | 1.21 | 13.84 | 2.56 |
| District X | 2557.15 | 27.32 | 1.07 | 27.32 | 1.07 | 81.78 | 3.20 |
| District XI | 952.80 | 0.14 | 0.02 | 0.14 | 0.02 | 1.11 | 0.12 |
| District XII | 1845.12 | 377.76 | 20.47 | 393.44 | 21.32 | 298.46 | 16.18 |
| District XIII | 2563.52 | 1739.61 | 67.86 | 1758.71 | 68.61 | 1673.58 | 65.28 |
| District XIV | 1224.16 | 727.01 | 59.39 | 728.17 | 59.48 | 534.56 | 43.67 |
| District XV | 558.31 | 29.75 | 5.33 | 29.75 | 5.33 | 34.42 | 6.16 |
| District XVI | 369.45 | 32.04 | 8.67 | 32.04 | 8.67 | 60.45 | 16.36 |
| District XVII | 2378.69 | 25.80 | 1.08 | 23.93 | 1.01 | 40.58 | 1.71 |
| District XVIII | 6533.28 | 1789.34 | 27.39 | 1808.39 | 27.68 | 1575.99 | 24.12 |
| Cracow | 32,642.96 | 6879.47 | 21.07 | 7163.81 | 21.95 | 5742.80 | 17.59 |

The following listings will include only the most dangerous scenario for Cracow, i.e., BPs for the Tresna and Porąbka reservoirs disaster (highest culmination in Cracow and the largest flooded area—Figures 7c and 8).

Table 7 shows the depth structure of the inundated areas of Cracow's administrative units, in 4 depth ranges (0–0.5 m, 0.5–2.0 m, 2.0–4.0 m, and above 4.0 m), which is mainly due to the natural topography within the administrative units, as well as artificial obstacles to water flow in the form of buildings.

**Table 7.** List of flood inundations in Cracow with a breakdown into depth zones for the Tresna and Porąbka disasters, broken down by subsidiary districts.

| Administrative Unit | Total Area of the Flood Lagoon (ha) | Including Depth (ha): | | | |
|---|---|---|---|---|---|
| | | 0–0.5 m | 0.5–2.0 m | 2.0–4.0 m | above 4 m |
| 1 | 2 | 3 | 4 | 5 | 6 |
| District I | 174.54 | 25.88 | 108.99 | 33.60 | 6.07 |
| District II | 327.20 | 20.66 | 104.76 | 193.77 | 8.01 |
| District III | 2.63 | 1.22 | 1.41 | 0.00 | 0.00 |
| District IV | 8.77 | 2.35 | 6.42 | 0.00 | 0.00 |
| District V | 189.50 | 37.39 | 121.74 | 30.37 | 0.00 |
| District VI | 6.37 | 1.98 | 4.31 | 0.08 | 0.00 |

**Table 7.** *Cont.*

| Administrative Unit | Total Area of the Flood Lagoon (ha) | Including Depth (ha): | | | |
|---|---|---|---|---|---|
| | | 0–0.5 m | 0.5–2.0 m | 2.0–4.0 m | above 4 m |
| 1 | 2 | 3 | 4 | 5 | 6 |
| District VII | 490.70 | 13.54 | 118.33 | 143.03 | 215.79 |
| District VIII | 1155.66 | 72.69 | 212.86 | 460.12 | 409.98 |
| District IX | 6.56 | 1.28 | 4.75 | 0.53 | 0.00 |
| District X | 27.32 | 14.37 | 10.08 | 2.87 | 0.00 |
| District XI | 0.14 | 0.00 | 0.09 | 0.05 | 0.00 |
| District XII | 393.44 | 46.49 | 123.88 | 218.15 | 4.93 |
| District XIII | 1758.71 | 101.24 | 633.47 | 900.94 | 123.06 |
| District XIV | 728.17 | 24.82 | 250.63 | 394.49 | 58.23 |
| District XV | 29.75 | 5.85 | 19.58 | 4.32 | 0.00 |
| District XVI | 32.04 | 8.89 | 21.01 | 2.14 | 0.00 |
| District XVII | 23.93 | 10.78 | 12.66 | 0.49 | 0.00 |
| District XVIII | 1808.39 | 307.70 | 716.20 | 499.32 | 285.17 |
| Cracow | 7163.81 | 697.15 | 2471.17 | 2884.26 | 1111.24 |

*4.2. Flood Risk in Cracow from Disasters of Hydraulic Structures*

The following Table 8 lists the magnitude of the flood risk (damage and losses) for the Cracow area in the structure of the administrative units. Table 8 column 2 shows the structure of the flooded areas for the worst-case scenario (failure of the Tresna and Porąbka dams) broken down by land use categories. Table 8 column 3 illustrates the magnitude of flood losses calculated using Formulas (1) and (2) based on per-unit rates from column 4 of Table 3, disregarding the water and other land category for which losses are not calculated according to the methodology [30].

Table 8 columns 5–10 shows the number of residents and public facilities at risk classed into 6 categories:

- Scientific and educational institutions (schools, kindergartens, nurseries, school and education centers, universities),
- Cultural and sports facilities (monuments, museums, cinemas, theaters, community centers, libraries, halls, stadiums, sports fields),
- Health care and social welfare facilities (hospitals, clinics, sanatoriums, social care homes, nursing homes, hospices),
- Uniformed services establishments (police stations, border guard stations, fire stations, prisons, detention centers, correctional facilities),
- Religious buildings (churches, chapels, religious houses, parish houses, seminaries),
- Commercial and tourist facilities (shopping centers, hotels, holiday homes, camping sites).

As can be seen from Table 6, the most serious threat to the city is posed by the BP scenario for the Tresna and Porąbka dam disaster which floods over 7163 ha in Cracow, i.e., almost 22% of the city area. The Goczałkowice disaster will flood 6879 ha in Cracow, which is 21% of the city area, while the smallest inundation is provided for by scenario I—5743 ha and 17.6% of the city's area. Of Cracow's districts, those located along the Vistula axis are obviously the most at risk, including District XIII, where nearly 70% of its area is flooded.

Within the city, shallow flooding up to 0.5 m occurs in 9.73% of the area, flooding between 0.5 and 2.0 m in 34.5% of the area, flooding between 2.0 and 4.0 m in 40.26% of the area, and the deepest flooding occurs in 15.51% of the area. The largest area of the most dangerous zones (over 4.0 m deep), which covers an area of 1111.24 ha in Cracow, is in District VIII (nearly 410 ha); half of Cracow's Districts—III, IV, V, VIIX, X, XI, XV, XVI, and XVII do not suffer such deep floods. The average depth of flood inundation in the city area is 2.44 m for the variant under consideration.

As far as the entire city is concerned, grassland, residential areas and agricultural land and permanent crops dominate in terms of flooded area, with 85%, 32%, and 44.5% of the

total area, respectively. The largest area of flooded residential grounds is in Districts XIII, VIII, XVIII, II, XIV, and I, industrial areas in Districts XIII, XII, and XIV, and communication areas in XIII, XIV, and II.

The overall flood damage in Cracow was estimated at PLN 31 billion. Of all Cracow's districts, District XIII suffers the most serious losses, followed by District VIII. In terms of losses, three categories of use dominate: residential, industrial and communication areas due to the extensive flooded areas and high per-unit loss rates. These 3 categories account for 99.9% of all losses in the city (Figures 9 and 10).

**Table 8.** The structure of flood risk in Cracow in terms of the flood area, flood losses, residents, and public facilities at risk—Tresna and Porąbka disaster scenario.

| Administrative Unit | Total Area of the Flood Lagoon (ha) | Total Amount of Flood Losses (PLN million) | Number of Inhabitants at Risk (people) | Number of Public Facilities at Risk (pcs) | | | | | |
|---|---|---|---|---|---|---|---|---|---|
| | | | | Scientific and Educational Institutions | Cultural and Sports Facilities | Health and Social Care Facilities | Uniformed Services Establishments | Religious Buildings | Commercial and Tourist Facilities |
| **1** | **2** | **3** | **4** | **5** | **6** | **7** | **8** | **9** | **10** |
| District I | 174.54 | 2057.99 | 53,823 | 94 | 43 | 46 | 6 | 44 | 317 |
| District II | 327.20 | 3484.55 | 54,636 | 36 | 10 | 32 | 9 | 5 | 179 |
| District III | 2.63 | 0.16 | 0 | 0 | 0 | 0 | 0 | 0 | 0 |
| District IV | 8.77 | 0.13 | 0 | 0 | 0 | 0 | 0 | 0 | 0 |
| District V | 189.50 | 1282.84 | 14,774 | 92 | 22 | 8 | 0 | 0 | 173 |
| District VI | 6.37 | 1.58 | 4 | 0 | 0 | 0 | 0 | 0 | 2 |
| District VII | 490.70 | 1640.48 | 14,869 | 16 | 32 | 18 | 1 | 5 | 304 |
| District VIII | 1155.66 | 6271.77 | 33,056 | 32 | 8 | 10 | 3 | 16 | 1079 |
| District IX | 6.56 | 2.14 | 0 | 0 | 0 | 0 | 0 | 0 | 0 |
| District X | 27.32 | 3.27 | 2 | 0 | 0 | 0 | 0 | 0 | 1 |
| District XI | 0.14 | 0.02 | 0 | 0 | 0 | 0 | 0 | 0 | 0 |
| District XII | 393.44 | 1119.12 | 5644 | 1 | 2 | 1 | 0 | 0 | 51 |
| District XIII | 1758.71 | 8115.51 | 52,612 | 56 | 23 | 19 | 5 | 11 | 1505 |
| District XIV | 728.17 | 3369.47 | 18,137 | 7 | 4 | 1 | 0 | 0 | 316 |
| District XV | 29.75 | 8.15 | 11 | 0 | 0 | 0 | 0 | 0 | 3 |
| District XVI | 32.04 | 153.50 | 130 | 0 | 1 | 0 | 0 | 0 | 23 |
| District XVII | 23.93 | 73.40 | 88 | 0 | 1 | 0 | 0 | 0 | 36 |
| District XVIII | 1808.39 | 3429.09 | 2938 | 6 | 3 | 2 | 0 | 5 | 701 |
| Cracow | 7163.81 | 31,013.17 | 250,724 | 340 | 149 | 137 | 24 | 86 | 4690 |

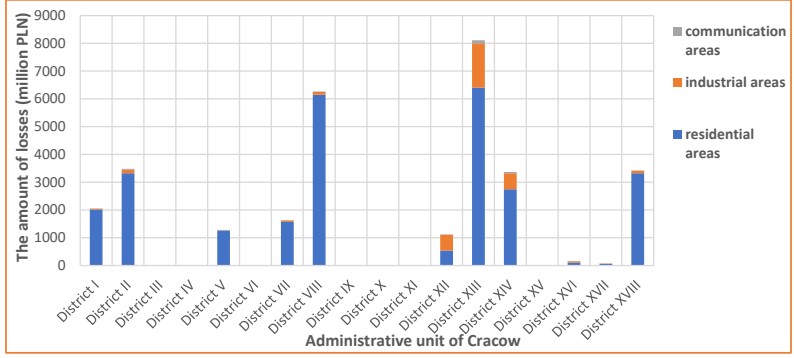

**Figure 9.** Tresna and Porąbka disasters—structure of losses by land use categories in Cracow.

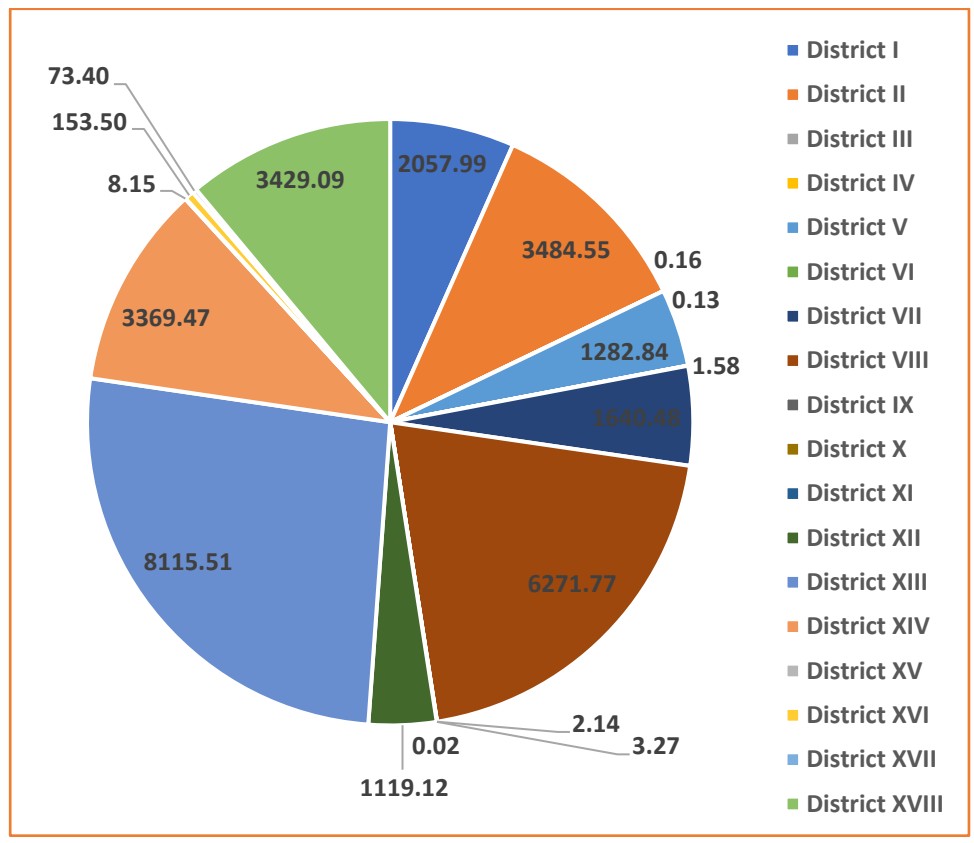

(**a**) Total losses by district

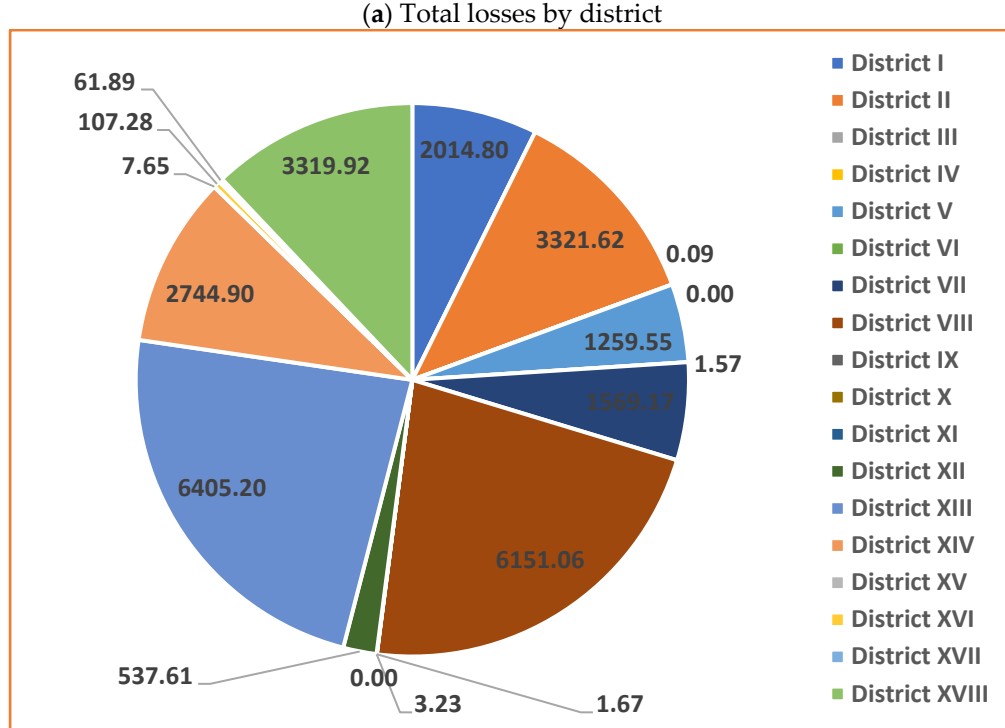

(**b**) Residential losses by district

**Figure 10.** *Cont*.

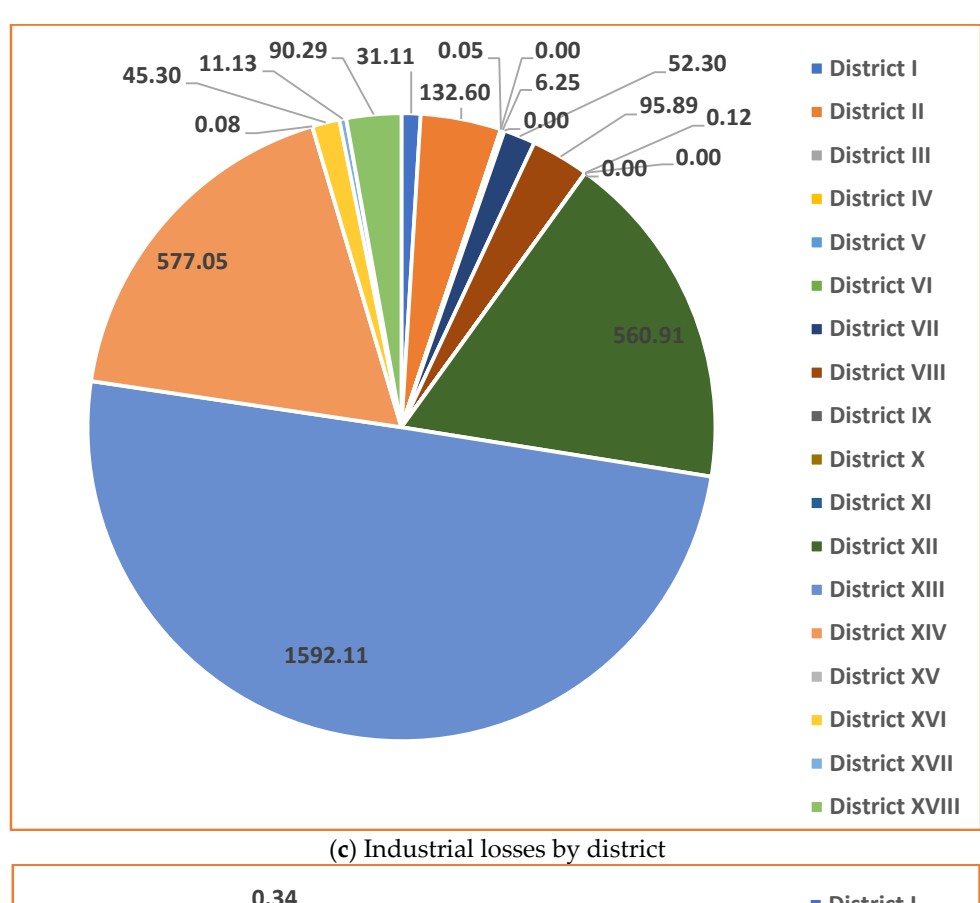

(**c**) Industrial losses by district

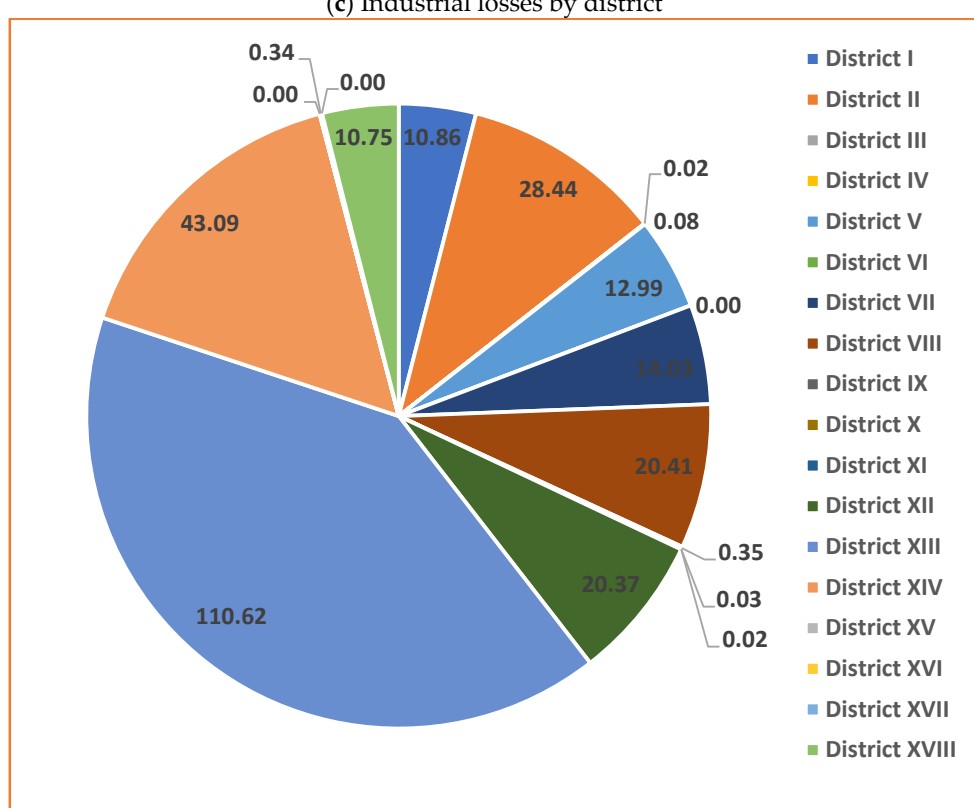

(**d**) Communications losses by district

**Figure 10.** Tresna and Porąbka disaster—total losses (**a**), residential losses (**b**) industrial losses (**c**) and communication losses (**d**) by district (in PLN million).

The number of inhabitants at risk in Cracow is over 250,000 (32.8% of the city's population), with the largest numbers in Districts II, I, XIII, and VIII. Of this number, approximately 12,800 people are likely to be in areas with the greatest depths, over 4 m—most of them in District VIII. Fewer than 5 people are at risk in Districts VI and X, and there are no people at risk in Districts III, IV, IX, and XI.

In total, more than 4500 public buildings are at risk of flooding across the city, mostly commercial and tourist facilities, educational and cultural institutions. Districts XIII and VIII have the highest number of buildings at risk. In the UNESCO-listed Old Town, a small part of Planty Krakowskie near the Cracow Philharmonic is at risk of flooding, while in the remaining area, a number of important buildings are at risk, including the National Museum, the Jagiellonian Library, the University of Agriculture, the AGH University of Science and Technology, as well as the Wisła and Cracovia stadiums and sports halls.

## 5. Discussion

The index method of calculating potential flood damage and losses used in Poland in the work on developing the MZP and MRP in the second planning cycle was used in this paper to estimate the magnitude of flood risk in Cracow. It has been partly modified by the author with regard to the per-unit loss index for residential areas and takes into account the city's spatial development and GDP data from 2016 (rather than data for the Małopolska voivodship). The value of this index is €582.2/m$^2$ for Cracow and 5 times the value of the indicator for the Małopolska voivodship, which may raise doubts as to whether it is not too high. For comparison, a similar index of housing losses (but with the added value of car losses) calculated for Cologne by Grünthal et al. [2] was at €1015/m$^2$ in 2006, i.e., 74% higher, from which one could conclude that the index for Cracow was not overestimated. Despite this convergence of results, however, there are arguments pointing to a possible overestimation of the size of this indicator. This is because the adopted method includes households among the most affected by direct losses. The method treats households in multi-story buildings in the same way—those located on the two lowest floors (where water enters the dwellings) as well as those located above the first floor (which do not suffer direct flooding of property, except for property stored in cellars and cars left in parking lots). This group should be excluded from the calculation of losses using formula (1) and should be the subject of detailed analytical work aimed at developing separate formulas for it (in the next planning timeframe), taking into account a minimum of material losses to property, but suffering indirect losses resulting from, e.g., staying in isolation on the upper floors of a flooded building for an extended period of time. This is an important issue, given the dominant share of residential losses in the overall losses of the city.

The overall flood losses in Cracow estimated using the proposed method for the most adverse variant of the considered scenarios were at PLN 31 billion (including PLN 27.5 billion in residential areas), which corresponds to 2016 price levels. When we convert this amount using factors that account for the 2021 rate of inflation [50], we get a value of PLN 35.7 billion, which is equivalent to approximately five annual city budgets.

Despite the caveats formulated above, the proposed method of estimating flood losses in Cracow is correct. It is primarily intended to indicate the spatial diversity of the areas in terms of the magnitude of potential losses. The per-unit loss indices calculated for Cracow and its subdivisions (districts) should (despite the modification introduced) still be treated as an approximation of the actual figures. Overestimation or underestimation is, it seems, an unavoidable phenomenon which occurs in index analyses regardless of which data are used and how aggregated they are. For example, Thieken et al. [51] checked aggregated census and property data for the whole of Germany against a list of losses for two historical floods. They found that the algorithm tends to underestimate the actual data in some areas and overestimate them in others.

## 6. Conclusions

1. Currently, Cracow is not prepared for protection against extreme flood scenarios. The current state of securing the city with embankments allows for safe flows within the water limits of Q1% (2411 m$^3$/s), while the estimated culmination of the 0.2% wave is 3156 m$^3$/s and overflows the embankments in the center of Cracow with a layer of 27 cm. Against this background, the largest so far documented historical flood of 1813 (Figure 6) in Kraków had a culmination of about 3300 m$^3$/s [47,49], thus it exceeded the wave with the probability of occurrence of 0.2%.

Moreover, the city is not prepared for protection against floods caused by the disasters of the Goczałkowice and Tresna dams with Porąbka, which generate higher culminations than Q0.2% water, of the order of 3578–3615 m$^3$/s and overflow the embankments with a layer of 41 and 48 cm, respectively. Hope for a significant improvement in this situation is provided by the results of works on the concept of building a system of 10 polders above Kraków, which, according to calculations, may reduce the 0.2% flood wave in the center of Kraków by approximately 80 cm.

2. This paper attempts to estimate the size of flood damage and losses in the area of the city of Cracow in the case of extremely extreme flood scenarios of flooding caused by the failure of dam reservoirs on the Vistula and Soła located above the city. The implementation of each of these scenarios results in flooding of significant areas of the city. Flood risk areas determined as a result of flood course modeling for these variants were used [33] and compared with scenario I for the 500-year flood [35]. It was indicated that the most dangerous scenario for Cracow is the disaster scenario of the Tresna and Porąbka dams.

3. The average breach width adopted in the modeling of the Goczałkowice and Tresna disasters is the closest to the Froehlich formula [20], the other formulas either significantly overestimate or underestimate the breach width. This tendency was identical in the analyses carried out for all earth dams in Poland. The Macdonalds and Langridge-Monopolis formula [21], for example, often gave values exceeding the length of the entire dam. This confirms similar results obtained in the works of other authors [15,16,28], where the Froehlich formula was usually rated the best. In addition, it indicates that the averaging assumption adopted in the methodology is the right approach to the problem in the case of a large dispersion of empirical results.

4. In terms of flood risk, spatial layers of the flood ranges were analyzed for each scenario and the depth structure for four ranges (up to 0.5 m, 0.5–2.0 m, 2.0–4.0 m, and above 4.0 m) in the area of Cracow and each of its auxiliary units (districts). This analysis makes it possible to prioritize the areas most at risk.

5. Despite the potentially significant threat and damage that may occur in the area of Cracow, the risk of the disaster scenario of hydrotechnical structures protecting Cracow against natural floods is negligible. The dam disaster scenario combined with the simultaneous occurrence of a flood with a probability of 0.2% in the entire basin is, in principle, only theoretically possible. All structures are class I facilities and have been assessed as non-threatening by the state service for the safety of damming structures in the period 2015–2019.

**Funding:** This publication is based on the results of projects entitled. "Przegląd i aktualizacja map zagrożenia powodziowego i map ryzyka powodziowego" and "Opracowanie map zagrożenia powodziowego i map ryzyka powodziowego dla obszarów narażonych na zalanie w przypadku zniszczenia lub uszkodzenia budowli piętrzących—cz.2" ("Review and update of flood hazard maps and flood risk maps" and "Development of flood hazard maps and flood risk maps for areas exposed to flooding in case of destruction or damage of damming structures—part 2") completed in 2019–2022 at the request of the national water management company Wody Polskie, which owns the materials and results used. Project No.: POIS.02.01.00-00-0013/16, financed by the Cohesion Fund of the European Union.

**Institutional Review Board Statement:** Not applicable.

**Informed Consent Statement:** Not applicable.

**Data Availability Statement:** Not applicable.

**Conflicts of Interest:** The author declares no conflict of interest. The funding sponsors had no role in the design of the study; in the collection, analyses, or interpretation of data; in the writing of the manuscript, or in the decision to publish the results.

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
