# Peer review of "Assessment of the Potential Flood Hazard and Risk in the Event of Disasters of Hydrotechnical Facilities—The Exemplary Case of Cracow (Poland)"

_water, doi:10.3390/w15030403_

Round 1

Reviewer 1 Report

This paper presents an analysis of the potential consequences for Cracow of failure of dams in the Vistula catchment. The estimation of losses and damages have been made based on model studies and flood hazard and flood risk assessments. An adjustment of the loss index for Cracow's residential areas was proposed. The results may be interesting and reasonable. However, some improvements have to be made before it can be accepted for publication.

Major comments:

1. The chapter arrangement of the paper seems unreasonable. The part of the results is too simple. Only some figure and tables are illustrates, but no data analysis have been made. The part of Discussion isn't really a discussion, but a describe of the results. Actually, some discussions have been stated in the part of Summary and conclusions. Thus, I suggest the author to merge the two parts of Results and Discussion of the results into one part: Results; and change the part of summary and conclusions to be Discussion; and then add a part of Conclucions at the end of the paper. 

2. The part of discussion (line 509-590 in the present version) has to be reinforced. In the manuscript, the risk of flooding and losses have been estimated, and some comparison between different formula like the Froehlich, the Macdonald and Langridge Monopolis fomula. A basic assumption is that the river flood would not erode the river bed; so only the rise of water level was taken into account to estimate the river flooding, damage and losses. However, in some cases, the river bed change induced by river flood events would be significant and play important role in the water level changes during flood events. For example, both field data and numerical modeling in the Qiantang River Estuary of China found that, a river flood event can erode the river bed by up to 5 m and subsquently lower the water levels (Xie et al., 2017, 2018, 2021, 2022). Whether the river bed  can be eroded seriously depends on the peak river discharge during a flood event, and also depends on the sediment property in the river, namyly, fine sediments tend to be eroded more easily by river floods.

3. There are 9 tables and 13 figures in the manuscript. Some tables are quite similar and possibly can be merged into one table. You can also draw the data in the tables using figures. In my opinion, figures are more straight forward than tables to show information. 

Minor comments:

1. The writting is overall well, but there are some grammar and spelling errors. For example, line 22-23, two Venice[3}, and please change } with ]. line 26, please give the full name of UNESCO, ICCROM, ICOMOS, IUCN when the abbreviations first appear. line 44, I think the bullet is redundant here. lines 133 and 145, check "of or". line 373, the 3 of m3/s should be a superscript. Probably there are other writting errors. Please check through the paper carefully.

2. Figure 2 is difficult to understand. For instance, what is Max PP? what do you mean by different lengths of the white parts of the dams, what is the blue curves and arrows in the left of the panels. Please explain in the figure caption or in the context. 

3. Figure 6, the place names in this figure is in Polish. Please change them into English place names.

4. Figure2 7-9, the legends and words in these figure are too small and unclear, please revise. Possibly, these three figures can be merged into one figure which contains three panels a, b, c.

5. The format of the reference list is different from the format of this journal WATER, please check them. In addition, in the reference list, there are 29 references are in Polish that is very difficult to understand for readers from other countries. For a scientific paper published in an international pioneering Journal, it's better to cite more references in English. 

6. In the abstract, please add some sentences to state the value of this study to other systems in the world.

References:

1. Xie D., Wang Z. B., 2021. Seasonal tidal dynamics in the Qiantang Estuary: The importance of morphological evolution. Frontiers in Earth Science, 9, 782640, doi: 10.3389/feart.2021.782640.

2. Xie D., Gao S., Wang Z. B., Pan C., Wu X., Wang Q., 2017. Morphodynamic modeling of a large inside sandbar and its dextral morphology in a convergent estuary: Qiantang Estuary, China. Journal of Geophysical Research: Earth Surface, 122, 1553-1572, doi: 10.1002/2017JF004293.

3. Xie D., Wang Z. B., Huang J., Zeng J., 2022. River, tide and morphology interaction in a macro-tidal estuary with active morphological evolutions. Catena, 212, 106131, doi: 10.1016/j.catena.2022.106131.

4. Xie D., Pan C., Gao S., Wang Z. B., 2018. Morphodynamics of the Qiantang Estuary, China: Controls of river flood events and tidal bores. Marine Geology, 406, 27-33, doi: 10.1016/j.margeo.2018.09.003.

Author Response

Please refer to the attachement

Reviewer 2 Report

Assessment of the potential flood hazard and risk in the event of disasters of hydro-technical facilities.  The exemplary Case of Cracow (Poland)

Abstract: short, clear, and concise.

Introduction: Introduction provides a good review of the research articles related to the study.

Methodology: How the sentences are framed is not very clear. Better Formation may provide clarity.

Results: The figures used in the result section are not clear at zoom, due to which the legend is not clear to understand the area better.

While using extensive tabular data, a better description would provide clarity.

Summary and Conclusion: The concluding remarks of the study are not clear.

A major focus is based on the possibility of property damages in terms of value, whereas the conclusion lists that the risk of a disaster scenario for cracow’s natural flood protection hydraulic structures happening is negligible.

The study needs a proper justification as to how it will be helpful for future scenarios or the need for the study.

Why calculate the risk if it is mentioned in the conclusion that the scenario of a dam disaster and a 0.2% probability flood happening simultaneously throughout the basin is in principle only theoretically possible?

Even after that if any flood event takes place what will be the strategies for redevelopment, and how the consequence can be limited?

“Nevertheless, it is worth noting that a reservoir with a certain amount of free flood reserve to protect Cracow against flooding with the reservoir full will discharge a volume of water 3-5 times the amount of that reserve in an uncontrolled manner into the channel downstream of it in the event dam failure.” Not clear.

The study provides a good holistic approach towards dam failure disasters resulting in flood but not enough actual events of floods provides a weak impression of the study.

Major contradictions are found in the conclusion of the study whereas the write-up provides significant-good literature.

Round 2

Reviewer 1 Report

The authors have treated almost all of my suggestions. I have no further suggestions or comments.

Reviewer 2 Report

All corrections accepted